# SEMANTIC APPROACH TO AGENT ROUTING USING A HYBRID ATTRIBUTE-BASED RECOMMENDER SYSTEM

**Motivation**: Traditionally contact centers route an issue to an agent based on ticket load or skill of the agent. When a ticket comes into the system, it is either manually analyzed and pushed to an agent or automatically routed to an agent based on some business rules. A Customer Relationship Management (CRM) system often has predefined categories that an issue could belong to. The agents are generally proficient in handling multiple categories, the categories in the CRM system are often related to each other, and a ticket typically contains content across multiple categories. This makes the traditional approach sub-optimal. We propose a Hybrid Recommendation based approach that recommends top N agents for a ticket by jointly modelling on the interactions between the agents and categories as well as on the semantic features of the categories and the agents.

**Model Architecture:** We have available a dataset of tickets categorized into 41 categories, which are in principle mutually exclusive. However, some categories are more related to each other and therefore we must account for sematic information in the model. We also have a dataset of interactions between agents and categories of tickets.

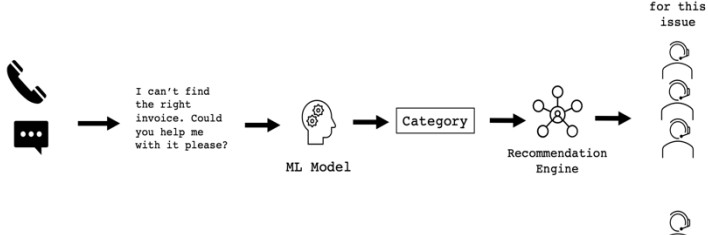

Figure1: Proposed Recommender System

To describe the model formally, let A be the set of Agents, C be the set of categories. Each Agent interacts with a number of categories, either favorably (a positive interaction), or unfavorably (a negative interaction). Each Agent and Category is represented by its meta data $F_A$: {Total Case Completion Time, Average Sentiment Score, Average Customer Rating} and $F_C$: $\left\{\frac{1}{N}\sum_{j=0}^{N}E_{ij}\right\}$, where N= Number of Tickets in a Category, E=Word2Vec semantic embedding [1] vector of dimension [d: 300]. The interaction dataset between the categories and the agents is represented by the count of tickets the agent has resolved for a particular category: $I = \{i_{a1}i_{c1}, i_{a2}i_{c2}, \cdots, i_{am}i_{cn}\}(a_m \in A, c_n \in C)$ and counts by R = $\{r_{a1,c1}, r_{a2,c2}, \cdots, r_{am,cn}\}(a_m \in A, c_n \in C)$ where m is equal to the total number of the agents and the n is equal to the number of the categories and $r_{am,cn}$ is the count of tickets solved by Agent $a_m$ for Category $c_m$. We formulate the recommendation problem using a hybrid attribute based model called LightFM[2] which is based on jointly factorizing the agent-category interaction matrix , category-feature, and agent-feature matrices. When a new ticket $(q_n)$ comes into the system we encode $q_n$ into a Word2Vec vector of dimension d: 300. We implemented Approximate Nearest Neighbor search (ANN) using Locality Sensitive Hashing [3] on the issue encodings to predict the category of the ticket $(q_n)$. Given different categories such as $c_1, c_2, ..., c_n \in R^d$, where d: 300 is the of dimension of the category encoding and a query point $q \in R^d$, top category is given by $Argmin\ [D(c_i, q)]$ where D is distance metric, Cosine Similarity. The hybrid recommender (Figure 1) calculates the similarity scores between the ticket and the categories to predict the category with the highest similarity score as the suggested category($c_p$). Category ($c_p$) is used to retrieve the top N agents from the recommender model.

**Model Evaluation**: We measure model accuracy using the mean receiver operating characteristics area under the curve (ROC AUC) metric. For a category, AUC corresponds to the probability that a randomly chosen positive agent will be ranked higher than a randomly chosen negative agent. We compute this metric for all categories in the test set and average it for the final score as shown in Table 1.

| Metric | logistic | BRP** | WARP** |
|---|---|---|---|
| Train AUC (no features) | 0.81 | 0.738 | 0.761 |
| Test AUC (no features) | 0.84 | 0.785 | 0.782 |
| Train AUC (with features) | 0.8235 | 0.75 | 0.773 |
| Test AUC (with features) | 0.8562 | 0.812 | 0.79 |

Table 1: AUC metric computed for different loss functions. We see that the

**Conclusion**: The proposed recommender system, which considers meta data features along with the historic interactions between agents and categories, is better when compared to the traditional approach of agent routing. This is of high importance when a ticket is of a new category or if the top recommended agent is busy. Our results also show that the model augmented by the meta data features outperforms the standard collaborative filtering model. This recommender system recommends top N agents who can best solve the problem despite their experience and background.

** Refer to [2] [10] for WARP and BPR loss functions

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

** Refer to [2] [10] for WARP and BPR loss functions