# OpenReview forum: "SEMANTIC APPROACH TO AGENT ROUTING USING A HYBRID ATTRIBUTE-BASED RECOMMENDER SYSTEM"
_ICLR.cc/2021/Conference — Reject_

### Official Review · AnonReviewer1 · 2020-10-28
**Semantic Approach to Agent Routing using a Hybrid Attribute-based  Recommender System**

**Rating:** 2
**Confidence:** 5

**Review:**

* First of all, the submission is not following the style.

* The paper only has one page.

* The proposed method is not novel, and it is not described clearly.

* Lack of experiments and comparisons. Similarly, a one-page paper will have a very hard time to deliver enough story. There is only one table in the whole paper.

* BRP or BPR?

* I also found that it is on a YouTube video (https://www.youtube.com/watch?v=Rj5E724O4nA&ab_channel=anwithaparuchuri) mentioning it is a submission to another venue (WeCNLP). It looks the results (the only table) are totally identical, so it probably should be considered as dual submission. Also from the video, probably it also violates the double-blind policy.

---

### Official Review · AnonReviewer4 · 2020-10-30
**The submission is a comparison of three existing algorithms on a task recommendation problem**

**Rating:** 2
**Confidence:** 5

**Review:**

The submissions deals with the problem of assigning tickets to agents in a Customer Relationship Management (CRM) system. The motivation is that the traditional way of assigning tickets does not take into account the interactions between agents and the categories of the tickets. Three algorithms (Table 1) are compared on the AUC measure, however the details of the experiment are not clear. It is also not clear how the experiments were implemented. Figure 1 is hard to read and the caption on Table 1 is incomplete. Several features are used for agents and tickets but their details are not provided, e.g., how are the word2vec features described in Section "Model Architecture" obtained? The results are hard to understand too. Which model are being compared exactly, and what are their parameters? The "Model Architecture" section talks about a LightFM based model, while the table describes a model named "logistic", another named "BRP" (is it supposed to be the well known BPR-MF algorithm?) and the third one named "WARP." None of these are described. The conclusion refers to a "proposed recommender system", but no such system is proposed in the paper. This submission cannot be accepted to the conference.

---

### Official Review · AnonReviewer2 · 2020-11-01
**Using lightFM to solve this application is interesting. However, there is no modification on lightFM, except testing three loss fuction. The theoretical contribution and novelty are limited.**

**Rating:** 3
**Confidence:** 5

**Review:**

This paper proposes to use lightFM to solve the matching of the agent and the ticket belonging to some categories.

Using lightFM to solve this application is interesting. However, there is no modification on lightFM, except testing three loss fuction. The theoretical contribution and novelty are limited.

The experiments are also not enough. They are lightFMs of three loss fuctions, without other compared algorithms. So the statements “is better when compared to the traditional approach of agent routing” and  “outperforms the standard collaborative filtering model” in the Conclusion section are not convincing.

---

### Decision · Program_Chairs · 2021-01-07
**Final Decision**

**Decision:**

Reject

**Comment:**

Does not seem to be a complete submission (only one page), all reviewers agree on rejecting.